# Follicular Helper and Regulatory T Cells Drive the Development of Spontaneous Epstein–Barr Virus Lymphoproliferative Disorder

**DOI:** 10.3390/cancers15113046

**Published:** 2023-06-03

**Authors:** Elshafa Hassan Ahmed, Mark Lustberg, Claire Hale, Shelby Sloan, Charlene Mao, Xiaoli Zhang, Hatice Gulcin Ozer, Sarah Schlotter, Porsha L. Smith, Frankie Jeney, Wing Keung Chan, Bonnie K. Harrington, Christoph Weigel, Eric Brooks, Haley L. Klimaszewski, Christopher C. Oakes, Tamrat Abebe, Muntaser E. Ibrahim, Lapo Alinari, Gregory K. Behbehani, Polina Shindiapina, Michael A. Caligiuri, Robert A. Baiocchi

**Affiliations:** 1Comprehensive Cancer Center, The James Cancer Hospital and Solove Research Institute, The Ohio State University, Columbus, OH 43210, USA; elshafa.ahmed@osumc.edu (E.H.A.); shelby.sloan@osumc.edu (S.S.); charlene.mao@osumc.edu (C.M.); sarah.schlotter@osumc.edu (S.S.); porsha.l.smith.civ@health.mil (P.L.S.); frankie.jeney@uchealth.org (F.J.); wing.chan@osumc.edu (W.K.C.); eric.brooks@osumc.edu (E.B.); christopher.oakes@osumc.edu (C.C.O.); lapo.alinari@osumc.edu (L.A.); polina.shindiapina@osumc.edu (P.S.); 2Division of Infectious Disease, Department of Internal Medicine, Yale University, New Haven, CT 06520, USA; mark.lustberg@yale.edu; 3Department of Biomedical Engineering, College of Engineering, The Ohio State University, Columbus, OH 43210, USA; hale.409@osu.edu; 4Department of Biomedical Informatics/Center for Biostatistics, The Ohio State University, Columbus, OH 43210, USA; xiaoli.zhang@osumc.edu; 5Department of Biomedical Informatics, The Ohio State University, Columbus, OH 43210, USA; ozer.9@osu.edu; 6Division of Hematology, Department of Internal Medicine, The Ohio State University, Columbus, OH 43210, USA; christoph.weigel@osumc.edu (C.W.); gregory.behbehani@osumc.edu (G.K.B.); 7College of Veterinary Medicine, Michigan State University, East Lansing, MI 48824, USA; harr1540@msu.edu; 8College of Medicine, The Ohio State University, Columbus, OH 43210, USA; haley.klimaszewski@osumc.edu; 9Department of Microbiology, Immunology, and Parasitology, School of Medicine Tikur Anbessa Specialized Hospital, College of Health Sciences, Addis Ababa University, Addis Ababa AB1000, Ethiopia; tamrat.abebe@aau.edu.et; 10Department of Molecular Biology, Institute of Endemic Diseases, University of Khartoum, Khartoum 11111, Sudan; mibrahim@iend.org; 11City of Hope National Medical Center, Duarte, CA 91010, USA

**Keywords:** Epstein–Barr virus (EBV), lymphoproliferative disorder (LPD), immunocompromised patients, risk factors, CD4+ T helper subsets, Hu-PBL-SCID model

## Abstract

**Simple Summary:**

Over 90% of the adult population worldwide is infected with the Epstein–Barr virus (EBV). While EBV infection is associated with the development of lymphoproliferative disorders (EBV-LPD) in people with weakened immune systems, only ~20% of immunodeficient individuals develop EBV-LPD. Such clinical heterogeneity may reflect host variables that increase the risk of developing EBV-LPD. Immunodeficient mice engrafted with blood cells from EBV+ individuals develop spontaneous EBV-LPD of human B-cell origin with similar heterogeneity observed in humans. Our study aimed to investigate differences between the model’s lymphoma producers (High-Incidence, HI donors) and non-lymphoma producers (No-Incidence, NI donors). HI donors showed high levels of T follicular helper (Tfh), regulatory T cells (Treg), and myeloid-derived suppressor cells compared to NI donors. Depletion of Tfh or Treg subsets delays or prevents EBV-LPD in this model. Our results reveal potential biomarkers that may help classify vulnerable patients at risk for developing EBV-LPD.

**Abstract:**

Epstein–Barr virus (EBV) is a ubiquitous herpes virus associated with various cancers. EBV establishes latency with life-long persistence in memory B-cells and can reactivate lytic infection placing immunocompromised individuals at risk for EBV-driven lymphoproliferative disorders (EBV-LPD). Despite the ubiquity of EBV, only a small percentage of immunocompromised patients (~20%) develop EBV-LPD. Engraftment of immunodeficient mice with peripheral blood mononuclear cells (PBMCs) from healthy EBV-seropositive donors leads to spontaneous, malignant, human B-cell EBV-LPD. Only about 20% of EBV+ donors induce EBV-LPD in 100% of engrafted mice (High-Incidence, HI), while another 20% of donors never generate EBV-LPD (No-Incidence, NI). Here, we report HI donors to have significantly higher basal T follicular helper (Tfh) and regulatory T-cells (Treg), and depletion of these subsets prevents/delays EBV-LPD. Transcriptomic analysis of CD4+ T cells from ex vivo HI donor PBMC revealed amplified cytokine and inflammatory gene signatures. HI vs. NI donors showed a marked reduction in IFNγ production to EBV latent and lytic antigen stimulation. In addition, we observed abundant myeloid-derived suppressor cells in HI donor PBMC that decreased CTL proliferation in co-cultures with autologous EBV+ lymphoblasts. Our findings identify potential biomarkers that may identify individuals at risk for EBV-LPD and suggest possible strategies for prevention.

## 1. Introduction

Epstein–Barr virus (EBV) belongs to the gamma herpes virus family and was the first human oncogenic virus to be described [1]. The virus is ubiquitous, with more than 90% of the world’s population becoming infected by young adulthood [2]. EBV demonstrates a narrow tropism for human epithelial cells and naïve B-lymphocytes. During primary infection, EBV-infected B-cells become activated and proliferate for a limited time, after which immunocompetent individuals mount an efficient adaptive, cellular immune response that usually controls infection. However, following primary infection, EBV maintains persistence within the host memory B-cell compartment, establishing life-long latency. The presence of immunosuppression (inherited, acquired, or due to iatrogenic treatment) compromises EBV-specific adaptive immunity. This may lead to immune escape of EBV-infected lymphocytes, B-cell immortalization, and polyclonal B-cell proliferation that has the capacity to evolve into malignant lymphoproliferative disorders (LPD) [3,4]. Additionally, the presence of EBV in diseases such as diffuse large B cell lymphoma (DLBCL) represents an independent adverse prognostic factor associated with poor response to therapy and shorter overall survival compared to EBV-negative DLBCL [5,6]. To our knowledge, there is no standard preventive or treatment approach for EBV-LPD, highlighting the need to better understand critical biomarkers to predict those at the highest risk. In cases of EBV-LPD in organ recipients (post-transplant lymphoproliferative disease, PTLD), a reduction in immunosuppression and treatment with rituximab may lead to immune reconstitution and resolution of the disease. This approach can be effective in the regression of PTLD in up to 30% of patients [7]. Yet, the risk of graft rejection is markedly increased [8,9].

While most individuals are EBV seropositive, only up to 20% of immunodeficient individuals will develop EBV-LPD, an observation that implies host factors may provide predisposing risk for EBV-LPD [7]. Several preclinical, spontaneous mouse models of EBV-LPD demonstrate similar findings [10,11,12,13]. In the human peripheral blood lymphocyte (PBL)-severe combined immunodeficiency (SCID) mouse model (Hu-PBL-SCID model), engraftment of SCID mice with PBMCs from EBV-seropositive individuals can lead to spontaneous EBV-LPD of human B-cell origin. EBV-LPD in this mouse model is reproducible, with phenotypic and genotypic similarities to LPD in humans. However, PBMCs from different human donors show different rates of EBV-LPD in engrafted mice [14]. Of all EBV-seropositive individuals, approximately 20% of donors will reproducibly induce EBV-LPD in 100% of engrafted mice (High-Incidence, HI donors). Intermediate/low-incidence donors develop EBV-LPD in up to 20–50% of mice, and No-Incidence donors (NI donors) in 0% of mice [14]. Furthermore, the development of EBV-LPD in the Hu-PBL-SCID model requires the presence of autologous CD4+ T-helper (Th) cells in the PBMC inoculum. Purified CD19+ mature B-cells injection into SCID mice results in poor engraftment and failure to produce human immunoglobulin or EBV-LPD [11,12,13,15].

Naïve Th cells, upon activation with antigen, differentiate into distinct populations, including T-helper 1 (Th1), Th2, Th17, regulatory (Treg), and T follicular helper (Tfh) cells. Each Th subset has distinct biologic and immunoregulatory properties (reviewed in references [16,17]. The signature cytokine for Th1 is IFNγ, and these cells are essential in protecting the host from infection with intercellular pathogens. Th2 cells help drive adaptive humoral immunity (antibody production) to protect against a wide variety of infections. Th17 cells promote protection from fungal infection and are associated with autoimmune diseases. Treg subsets express the IL2 alpha-receptor, CD25, and provide suppressive signals to regulate immune networks [18,19]. Treg cells are essential for suppressing autoimmunity and play a vital role in immune tolerance [20]. In the EBV infection setting, Tregs function as immune suppressive cells that can prevent CD8+ cytotoxic T lymphocytes (CTLs) from killing EBV-infected tumor cells [21]. Tfh cells are fundamental for germinal center (GC) formation, the primary site of B-cell maturation [22] and promote B-cell survival [23,24,25].

We hypothesized that distinct host immune profiles variables account for the high incidence of EBV-LPD phenotype in this preclinical model. We performed a detailed immunologic, immunophenotypic, and transcriptomic analysis of the peripheral blood mononuclear cell (PBMC) subsets from healthy donors. PBMCs from HI donors were found to have higher baseline levels of Tfh and Treg cells. HI donors also demonstrated significantly reduced T-cell memory to several latent and lytic EBV gene products. RNA-Seq expression profiles showed distinct pathways between NI and HI donors on gene set enrichment analysis (GSEA) on donors’ CD4+ Th cells collected ex vivo from engrafted mice. To evaluate the relevance of Tfh and Treg cells in EBV-LPD in vivo, we conducted a Th-selected subset depletion experiment confirming that Tfh and Treg subsets are critical drivers of EBV-LPD and act to promote lymphomagenesis in this preclinical model of EBV-LPD. HI donors also display high frequency of the suppressor cells, myeloid-derived suppressor cells or MDSCs. Our findings point toward novel potential biomarkers that may help identify patients at risk for developing EBV-LPD and offer insight into prevention or therapeutic approaches for this group of diseases.

## 2. Materials and Methods

### 2.1. Study Participants

The Ohio State University (OSU) Office of Responsible Research Practices and Institutional Review Board (IRB) approved this study. We recruited individuals for this study using soliciting flyers distributed throughout the OSU Columbus campus. Study participants provided informed consent using our IRB-approved protocol (1998H0240). EBV seroconversion of the study participants was confirmed by EBV viral capsid IgG test. EBV-negative individuals were excluded from further participating in this study. PBMCs were collected from the EBV+ donors and then enumerated and stored until used.

### 2.2. Mice

Institutional Animal Care and Use Committee approved all experimental animal procedures. Five to eight-week-old CB.17 scid/scid (SCID) mice were purchased from Taconic Farms (Germantown, New York, NY, USA) and housed in a pathogen-free environment. Animals showed no evidence of the “leaky” phenotype determined by flow cytometry, confirming the absence of B- and T-lymphocytes.

### 2.3. Donor Screening in the Hu-PBL-SCID

We inoculated PBMCs from EBV+ donors (fifty healthy human volunteers) into CB.17 SCID mice (5 × 10^7^ cells per mouse) as described previously [13,26]. Mice injected with human PBMCs were bled every other week. We centrifuged blood samples at 1000–2000× *g* for 10 min to collect the serum. The amount of human IgG in mouse serum was quantified to confirm engraftment using the Human IgG ELISA kit (eBioscience Inc., San Diego, CA, USA) per the manufacturer’s instructions. Engrafted mice were monitored for up to 120 days for lymphoma development. Then, donors were classified into HI and NI donors accordingly (Appendix A).

### 2.4. Quantitative Analysis of EBV DNA in Samples via qPCR

To quantify the number of EBV genome copies per donor B-cells, memory B-cells (CD19+/CD27+) were sorted using FACS Aria II (BD Biosciences, San Diego, CA, USA) from donor PBMCs. Total nucleic acid was extracted from sorted memory B-cells using QIAamp DNA Mini Kit (Qiagen, Germantown, MD, USA). Quantitative PCR (qPCR) was carried out with Fast SYBR green master mix (Applied Biosystems, Bedford, MA, USA) and run on a Viia7 qPCR machine (Applied Biosystems, Bedford, MA, USA). EBV DNA was quantified with primers specific to the EBV EBNA1 locus (forward: TCATCATCATCCGGGTCTCC, reverse: CCTACAGGGTGGAAAAATGGC), and signals were normalized to host genome DNA using primers specific for human beta-actin (ACTB) gene (forward: CAGGCAGCTCG-TAGCTCTTC, reverse: TTGTGGCTCAGGGAAAATGT). The relative gene expression was calculated as the 2-Δct method with Δct = ct(EBNA1) − ct(host genome control). We also quantified the EBV genome copies on mouse serum at multiple time points after PBMC inoculation.

### 2.5. Elispot Assay

Donor PBMCs (2 × 10^5^ cells) were stimulated with EBV latent antigens (EBNA-3A, EBNA-3C, and LMP1) pepmix (59 peptide pool of 15 mers with 11 amino acids overlap (JPT, Berlin, Germany)). PBMCs were also stimulated with the adenovirus encoding a recombinant full-length EBV lytic protein, BZLF1 (rAAV2/BZLF1) construct or control (rAAV2/null). BZLF1 is the immediate-early lytic cycle protein. We chose to study the BZLF1 protein for the following reasons: (1) BZLF1 is the first target antigen that is recognized by the immune system following primary infection [27,28]; (2) BZLF1 is a transcriptional activator that initiates the disruption of latency in EBV-infected cells [29]; (3) BZLF1 has been shown to promote B-cell transformation and lymphomagenesis [30,31]. The rAAV2/BZLF1 and rAAV2/null were constructed as described previously [32]. Secretion of human IFNγ was detected using Human IFNγ Enzyme-Linked Immunosorbent Spot (ELISpot) per the manufacturer’s instructions (Mabtech, Nacka Strand, Sweden).

### 2.6. Mass Cytometry

We developed a mass cytometry (Cytometry Flight of Time, CyTOF) panel of 33 metal-labeled monoclonal antibodies (Appendix A) for high-dimensional analysis of NI and HI donor PBMCs. Metal-labeled antibodies were purchased or conjugated in-house using the Maxpar antibody conjugation kit per the manufacturer’s instructions (Fluidigm, San Francisco, CA, USA) [33]. Cells were stained for the surface and intracellular antibodies as described previously [26]. Before data acquisition, cells were washed once with CSM and twice with sterile water, then resuspended in pure water mixed with 1:20 dilution of 4 elemental mass standard beads (Fluidigm) at 1 mL per million cells. Samples were collected on a Helios mass cytometer (Fluidigm) as described previously [34,35,36,37]. Normalized files [38] were uploaded to Cytobank software (Beckman Coulter, Brea, CA, USA)) for analysis [39,40,41].

### 2.7. Cell Sorting and RNA Isolation

Spleens were collected from mice four weeks after PBMC injection. Mouse spleens were prepared into single-cell suspensions then RBCs were lysed with 5–10 mL lysis buffer (Stem Cell Technology, Vancouver-Canada) for 10 min. Mouse splenocytes were enriched for human CD45+ cells using Miltenyi human CD45+ beads (Miltenyi Biotec, Cologne-Germany), according to the manufacturer’s recommendations. Then, human CD4+ Th cells were sorted to at least 95% purity from the human CD45+ enriched spleen using FACS Aria II (BD Biosciences, California-USA). Total RNA was isolated from sorted CD4+ Th cells using the Norgen kit per the manufacturer’s instructions (Norgen Biotek CORP, Ontario, BC, Canada).

### 2.8. Ampliseq Transcriptome Analysis

Gene expression profiling for ex vivo sorted CD4+ Th cells was generated using the Ampliseq transcriptome analysis described previously [42]. Briefly, ten nanograms of total RNA was reverse transcribed using a Super-Script Vilo cDNA synthesis kit. Barcoded adapters were added, and the Transcriptome libraries were generated using AmpliSeq primers and technology. Equal amounts from all libraries were combined and pooled for blend PCR using Ion Torrent. The number of reads affiliating to each gene target was represented as a count value. The normalized gene count data were imported into the R suite and analyzed using the DESeq2 software for differential expression. We also run the Gene set enrichment analysis (GSEA) [43] to detect significantly enriched gene sets. We ran the expression dataset of ex vivo sorted CD4+ Th cells from 3 HI donors and 3 NI donors against the Hallmark gene set collection from the Molecular Signatures Database (MSigDB) available within the GSEA software. We set-up the threshold at false discovery rate (FDR) < 25%.

### 2.9. Depletion of the Th Cell Subsets

PBMCs from 3 HI donors were used for this experiment. According to the manufacturer’s recommendations, the CD4+ Th cells were separated from PBMCs using Miltenyi human CD4+ beads (Miltenyi Biotec, Cologne-Germany). Then, CD4+ Th cells were stained with Biotin anti-human CXCR5 antibody or Biotin anti-human CD25 antibody (Biolegend, San Diego, CA, USA) to deplete Tfh and Treg, respectively using EasySep Biotin Positive Selection Kits (StemCell Technologies, Vancouver-Canada) with >90% purity. The CD4+ T-cells expressing the CD25 marker are highly enriched in Treg, and our CD25+ cell depletion primarily targets the FOXP3-positive CD4+ T-cells in donor PBMCs. Additionally, donors’ PBMCs were incubated with the beads for mock depletion and processed using the same procedures as the depleted counterpart cell preparations. Mock depleted PBMCs (positive control group), CD4+ Th-depleted PBMCs (negative control group), Tfh-depleted PBMCs (experimental group), or Treg-depleted PBMCs (experimental group) were injected via intraperitoneal (IP) route into 5–8-week-old SCID mice. Groups injected with Th-subset-depleted PBMC preparations received the equivalent numbers of non-depleted mononuclear cells as control animals (i.e., similar CD3/CD8, B cell, NK cell, monocyte/myeloid, and non-depleted CD3/CD4 subsets). Total number of PBMC preparations inoculated into mice ranged from 3.5 to 4.8 × 10^7^ cells per mouse. Appendix A provides additional data specifying the numbers of PBMC fractions used for engraftment. Mice were monitored and sacrificed when they showed signs of illness or at the end of this study (120 days).

### 2.10. Statistical Analyses

Linear mixed models were used to take account of the correlation among observations from the same animal engrafted with the same donor cells. Log-rank test was used for the survival analysis of the CD4 Th subset depletion experiment. Holm’s procedure was used to control for multiple comparisons. Two sample *t*-tests were used for independent data comparison, such as comparing immune responses between the NI donor and HI donor groups to EBV antigen stimulation.

## 3. Results

### 3.1. Identification of EBV-LPD High Incidence (HI) and No Incidence (NI) Donors

To identify select donors whose PBMCs induced spontaneous EBV-LPD in the Hu-PBL-SCID model, we inoculated SCID mice with PBMCs from a pool of healthy EBV+ donors. After PBMC inoculation, mice were monitored for signs of EBV-LPD, including weight loss, ruffled fur, inactivity, and palpable abdominal masses. We observed spontaneous lymphoma development in Hu-PBL-SCID mice 6–12 weeks after intraperitoneal injection of PBMC (from HI donors). EBV-LPD manifests as diffuse organ involvement (hepatosplenomegaly), anemia, and lymph node tumor burden. Tumor morphology is characterized as immunoblastic in nature with expression of type 3 EBV latency and expression of pan-human B-cell markers (CD19, CD20), similar to that seen with monomorphic PTLD [44]. They can be monoclonal, oligoclonal, or polyclonal, contain EBV DNA, and express lytic and latent EBV gene products [45].

Donors whose PBMCs’ reproducibly developed EBV-LPD in 100% of engrafted mice and donors whose PBMCs’ never led to the development of EBV-LPD in all engrafted mice, were classified as HI donors and NI donors, respectively (Appendix A).

### 3.2. Determination of EBV DNA Content for HI and NI Donors

When individuals are infected with EBV, the virus maintains latency in the memory B-cell compartment. To determine if differences in the EBV DNA viral load existed between groups, we measured the EBV genome copy number using a quantitative PCR (qPCR) on DNA isolated from memory B-cells sorted from HI and NI donor PBMCs. qPCR data confirmed no significant difference (*p*-value = 0.6629) in EBV DNA content in the B-cell compartment between the two groups (Figure 1a).

### 3.3. Recall Responses to EBV Antigens

We next evaluated memory adaptive immune responses to specific latent and lytic EBV antigens using PBMCs collected from HI and NI donors. We selected viral gene products that have been documented as well validated targets for adaptive T-cell immunity [46]. PBMCs were cultured in FBS-supplemented media containing pepmix for EBNA-3A, EBNA-3C, or LMP1. We performed the IFNγ ELISpot assay to determine immune responsiveness to pepmix stimulation. The number of IFNγ-secreting cells for NI donors was higher compared to HI donors (Figure 1b), and the difference reached statistical significance for EBNA-3A (*p*-value = 0.016) and EBNA-3C (*p*-value = 0.028). We also stimulated the PBMCs from NI and HI donors with an adenovirus preparation encoding recombinant full-length EBV lytic protein, BZLF1 (rAd-BZLF1), which showed a significant difference in the level of activation between the two groups (*p*-value = 0.0019) (Figure 1c). There were no variations in stimulation level between NI donors and HI donors when their cells activated with the empty adenovirus vector (rAd-null, *p*-value = 0.8970) (Figure 1c). These findings indicate that NI donors have a more robust recall response to EBV antigen than HI donors and support the existence of inherent immune-related differences between these groups.

### 3.4. Characterization of Immune Cell Subsets in NI and HI Donors

We next performed a detailed immunophenotypic characterization of PBMCs collected from NI and HI donors’ peripheral blood by mass cytometry. Samples were analyzed by unsupervised viSNE clustering on an equal number of CD45+ cells. The viSNE analysis is a dimensionality reduction visualization tool based on the t-distributed stochastic neighbor embedding (t-SNE) analysis that reduces the multidimensional distance cell events to a two-dimensional map [39]. The viSNE maps for NI and HI donors PBMCs were colored by channel, showing different lineage antigen expression levels (Figure 2a). We simultaneously view all lineage subsets using the viSNE overlaid tool with manual gating (Figure 2b). Appendix A shows detail of the gating scheme. T-cells from NI donors exhibited a 2-fold increase in CD3 mean expression compared to HI donors (*p*-value = 0.0006), while the mean expression of CD33 (myeloid cells) was higher among HI donors (*p*-value = 0.0410). No differences were observed for the CD19 (B-cells), and CD56 (NK cells) mean expression (*p*-value = 0.0742 and 0.0935, respectively) (Figure 2c). Subsequently, we computed the percentage of each mononuclear cell lineage in the CD45 gate from NI and HI donor PBMCs to compare the relative abundance between these two groups. The frequency of T-cells in the CD45+ gate was significantly greater in PBMCs from the NI donors (*p*-value = 0.0038) than in HI donors (Figure 2d). The mass cytometry analysis also revealed a unique feature in HI donors where the percentage of myeloid cells was significantly higher than those from NI donors (*p*-value = 0.0018) (Figure 2d). There was no significant difference in the frequency of B-cells (*p*-value = 0.2251) and NK cells (*p*-value = 0.05) between NI and HI donor PBMCs (Figure 2d).

We further investigated the T-cells and myeloid cells among the two groups. The percentage of the T-cell subset populations (CD4+ T helper (Th), CD8+ T-cells (CTLs), double-positive T-cells (CD4+/CD8+, 4/8 DP) and double-negative T-cells (CD4−CD8−, 4/8 DN) within the T-cell compartment were similar between NI and HI donors (Appendix A). We and others have previously reported the absolute need for Th in engrafted PBMC to generate spontaneous EBV-LPD in the Hu-PBL-SCID model [11,12,13,15]. Therefore, we further examined the CD4+Th subsets by running a FlowSOM unsupervised clusters on CD4+ viSNE clustering (Figure 3a). The unsupervised FlowSOM metaclusters of the analysis were indicated by the background color of FlowSOM nodes (example grid, Figure 3b). The size of the nodes depends on the number of cells represented by the particular node in the FlowSOM grid. The FlowSOM grid with the percentage of cells falling in the original manual gates (CD4+ Th gating scheme shown in Appendix A) showed that NI donor and HI groups displayed distinct patterns across the FlowSOM clusters (Figure 3b). The Th naïve cells (CD4+/CD45RA+) in FlowSOM clusters 1, 2, and 3 were abundant in NI-donor PBMC samples compared to HI donors (Figure 3b). However, the difference was not statistically significant (*p* = 0.1566) (Figure 3c). In contrast, the Th subset populations from individuals in the HI donor group were enriched compared to NI donors (Figure 3b) with the Tfh (CD3+/CD4+/CXCR5+/CD45RO+) and the Treg (CD3+/CD4+/CD25+/CD127-/FOXP3+) Th subsets were significantly more abundant in HI donor PBMCs compared to NI donor PBMCs (*p*-value = 0.0002 and 0.0307, respectively). The average percentage of the Treg for HI donor CD4+ T-cells is 9.8% and only 2.7% for NI donors. Likewise, the average rate of Tfh cells is 14.58% among HI donors and 3.8% among NI donors CD4+ T-cells (Figure 3d). Th1, Th2, and Th17 subsets were equally represented among NI and HI donors (*p*-value = 0.9522, 0.4552, and 0.3817, respectively) (Figure 3d). Further analysis of the CD33 myeloid cells confirmed the presence of large numbers of myeloid-derived suppressor cells or MDSCs (Lineage-/CD33+/CD11b+/HLA-DR low, gating scheme in Appendix A) among HI donors (*p* = 0.025) (Figure 3e). To examine the MDSCs inhibitory mechanism on CTLs, T-cells from NI and HI donors were activated with autologous lymphoblastoid cell lines (LCL) and cultured with sorted myeloid cells at a 1:1 ratio. Samples from the co-culture were collected on days 3, 5, and 7 for flow analysis. Higher rates of CD8+ cytotoxic T-cell (CTL) proliferation were observed in co-cultures established with PBMC from NI donors compared to HI donors, with statistical significance at day 7 (*p*-value = 0.0017) (Figure 3f). Most human cells recovered from the mouse spleen four weeks after PBMC engraftment were CD3+ T cells (Appendix A), consistent with previous literature [15,47]. We observed engraftment of both human CD4+ T-cells and CD8+ T-cells with no difference in T -cell subsets frequency between the HI and NI groups (Appendix A). Mass cytometry data of mouse splenocytes also showed engraftment of human B-cells. The B cell prevalence increased more than 7-fold in HI- compared to NI donors (Appendix A). At four weeks post-engraftment, myeloid cells are not detected. Tregs and Tfh retain the same pattern, but there are changes in their overall proportion. However, both are significantly elevated in HI compared to NI four weeks after engraftment.

### 3.5. CD4+ Th Cell from HI Genetic Signature: Correlated with Treg and Tfh Expansion and the Risk for the Development of EBV-LPD

Next, we evaluated the transcriptomic signature of ex vivo CD3+/CD4+ T cells (Th cells) purified from the spleens of Hu-PBL-SCID mice. We hypothesized that the crosstalk between B-cells and Th subsets in the chimeric environment occurs early and promotes the development of EBV-LPD. To capture gene expression signatures before the development of EBV-LPD, we inoculated SCID mice with PBMCs from HI and NI donors. We sacrificed mice four weeks after PBMCs inoculation, collected spleen, and sorted for human CD4+ cells ex vivo to >95% purity. RNA was isolated from sorted human CD4+ Th cells and then analyzed for differentially expressed genes (DEGs) by AmpliSeq Transcriptome analysis. The top DEGs depicted on the unsupervised heatmap clustering separated the genes that are relatively overexpressed and reduced in HI Th cells (Figure 4a). We compared the gene expression between the HI and NI donors using the Gene Set Enrichment Analysis (GSEA) software [43]. We identified significant upregulation of many hallmark gene sets [48] in HI donors, including the TNFA Signaling via NFKB, IL6/JAK/STAT3 Signaling, and TGF-beta signaling pathways (Figure 4b). The expression of critical genes involved in Treg (*JAK*/*STAT*, *TGFBR2*, *SMAD*, *FOXP3*) and Tfh (*IL6R*, *JAK1*, *STAT3*) polarity in these Hallmark gene sets is represented in Figure 4c. Additionally, we found 17 Hallmark gene sets significantly enriched in HI donors at false discovery rate (FDR) < 25%. These gene sets involved cell signaling, immune response, metabolic pathway, proliferation, and DNA damage (Figure 4d). Fewer Hallmark gene sets were significantly upregulated in NI donors, but none reached the FDR < 25% threshold (Figure 4d). The Th cell transcriptional program broadly differs between HI and NI donors.

### 3.6. Depletion of Tfh and Treg: Delays/Prevents EBV-LPD and Improves Survival in Hu-PBL-SCID Mice

Given that CD4+ Th cells in PBMCs are essential for the development of EBV-LPD, the primary question for this experiment was whether depletion of specific CD4+ Th subsets found expanded in HI donors will reduce lymphomagenesis and prolong survival in the Hu-PBL-SCID model. Because we only detected statistically significant differences in Tfh and Treg subsets, we focused on the depletion of these subsets to test if the outcome of the EBV-LPD rate would differ. PBMCs from 3 HI donors were depleted of Tfh cells or Treg cells using Biotin-conjugated antibody against CXCR5 and CD25 on CD4+ Th cells (detailed in materials and methods section and Appendix A). Mock depleted PBMCs and CD4-depleted PBMCs served as positive and negative controls, respectively. We measured the human IgG levels in mouse serum samples to compare engraftment efficiency. Overall, there was no difference in engraftment efficiency measured by human IgG levels in mouse serum among the experimental groups and PBMCs (~0.1 mg/mL by week 6) (Figure 5a). As expected, human IgG levels for the negative control group, CD4-depleted PBMCs, were significantly lower (*p*-value ˂ 0.0001) than the PBMCs group (Figure 5a). We also quantified EBV genome copies in mouse serum using an *EBNA1* qPCR assay. The EBV genome copies increased over time, but there were no differences between the PBMCs and experimental groups (Figure 5b). The H&E staining of the spleen and liver from mice engrafted with CD4-depleted PBMCs showed no evidence of EBV-LPD and was similar to the tissues from mice with no human cell engraftment. No tumor burden was detected among the CD4-depleted PBMCs and un-engrafted mice (Appendix A). In contrast, numerous neoplastic lymphocytes were found to efface the spleen and liver in mice from the mock depleted PBMCs group and mice with EBV-LPD from the experimental groups (Appendix A). However, several mice in the Tfh-depletion group developed less invasive disease with fewer lesions in the mouse liver (Appendix A).

We used the log-rank test to compare the survival of animals engrafted with a specific type of CD4+ Th depletion to the positive control group. After adjustment for multiple comparisons with Holm’s procedure, the results showed that Treg depletion significantly improved survival compared to the PBMC group (*p*-value = 0.01). The Tfh-depleted cohort also had improved survival but did not reach significance (*p*-value = 0.08) (Figure 5c). Depletion of Tfh and Treg depletion improve survival.

## 4. Discussion

EBV is a B-lymphotropic gamma herpes virus that infects more than 90% of the world’s population and is associated with several malignancies, including aggressive lymphoproliferative disorders (LPD) in immunocompromised individuals. EBV-LPD represents a significant complication encountered with multiple high-risk patient populations, including solid organ and stem cell transplantation recipients and immunodeficient states, and is associated with a poor prognosis. There is no standard approach to prevent or treat EBV-LPD, and the outcomes vary dramatically between institutions that use different treatment protocol. Importantly, no method exists to predict which individuals are at the higher risk for EBV-LPD.

Despite the ubiquitous nature of EBV, less than 20% of EBV-seropositive immunocompromised patients will develop EBV-LPD. Such clinical heterogeneity may reflect the presence of host variables that predispose individuals to EBV-LPD. Indeed, we have identified several host variables in this study that we believe may account for increased risk for developing EBV-LPD. These include reduced antigen recall of memory T cells in PBMC (Figure 1), expanded populations of absolute numbers of myeloid cells with myeloid-derived suppressor cell phenotype, and expansion of Treg and Tfh cells in the peripheral blood of HI donors.

The Hu-PBL-SCID mouse provides a preclinical, spontaneous model of EBV-LPD that shows immunophenotypic and viral latency (Latency III) to that seen in monomorphic EBV+ PTLD and some EBV+ diffuse large B cell lymphomas [49,50,51]. Moreover, this xenograft mouse model reflects similar heterogeneity between individual donors and their relative risk of generating EBV-LPD. Similar to the 20% incidence rate of LPD in vulnerable patient populations (solid organ transplant recipients), approximately 20% of healthy EBV-seropositive donors generate spontaneous EBV-LPD in the Hu-PBL-SCID model [14].

The tumor microenvironment is critical for efficient lymphomagenesis in the Hu-PBL-SCID model. Unlike other patient-derived xenograft models, the Hu-PBL-SCID model of EBV-LPD possesses autologous immune subsets that can drive [12,13] or prevent lymphomagenesis [26,52]. The ability of PBMCs from healthy EBV-seropositive humans to induce EBV-LPD in SCID mice requires the presence of autologous CD4+ T-cells in the PBMC inoculum [12,13,53]. Injection of purified B-cells alone is unsuccessful in producing EBV-LPD in SCID mice [53]. In vitro and in vivo treatments with anti-CD3 immunotoxin (anti-CD3 monoclonal antibody coupled to deglycosylated ricin A) to T-cell-deplete chimeric mice significantly reduced lymphoma development in the model [54].

Ma et al. illustrated the importance of the CD4+ Th compartment in supporting the B-cell lymphomagenesis in the cord blood mononuclear cell model where the mice were infected with an LMP-1-deficient EBV strain [11]. The critical role of CD4+ Th cells was also reported with patient-derived xenograft (PDX) in several histologic subtypes of B-cell lymphoma [11,55]. Burack and others demonstrated that CD4+ Th cells are essential tumor microenvironment elements to the PDX models, and depletion of CD4+ Th cells but not CD8+ T cells prevented PDX engraftment [55].

We hypothesized that HI donors identified in the hu-PBL-SCID model would display a distinct CD4+ Th immunologic and genetic signature associated with the development of EBV-LPD. To address this hypothesis, we utilized the Hu-PBL-SCID model to identify biomarkers that could identify individuals at risk for developing EBV-LPD. The current study showed that HI donors have higher Tfh, Treg and MDSC basal levels in peripheral blood, and the depletion of Tfh or Treg delayed/prevented EBV-LPD and enhanced survival.

Tfh cells are recognized as a distinct subset of CD4+ Th cells. Tfh cells drive the germinal center (GC) reaction, deliver essential growth factors for B-cell activation [23], promote B-cell differentiation in vitro [23,56]. In the EBV setting, Tfh cells are significantly increased in patients with primary EBV infection [57] and infectious mononucleosis [58] and are associated with the development of EBV+ lymphoma [25]. In our study, Tfh depletion delayed the onset of lethal EBV-LPD and, in some mice, prevented lymphomagenesis. These observations may account for the Tfh cell’s ability to increase the proliferative capacity of EBV-infected B-cells. Transcriptomic studies detected the hallmark *IL6*/*JAK*/*STAT3* pathway to be significantly upregulated from ex vivo purified CD4 T cells from HI donors (Figure 4b). IL6 activates the JAK/STAT3 pathway, an essential feature for Tfh differentiation. IL-6/JAK/STAT3 signaling regulates the ability of naive T cells to acquire B-cell helper capacity, and activation of the IL-6/STAT3 pathway leads to the secretion of Tfh signature cytokine, IL-21, which further enhances B-cell activity [59]. In addition, the IL6/JAK/STAT3 pathway was reported to be upregulated in EBV-associated epithelial cancers such as Nasopharyngeal carcinoma, gastric carcinoma, and oral squamous cell carcinoma [60].

Tfh cells are also characterized by high expression of CD40 ligand (CD40L), a co-stimulatory molecule that supports B-cell function and maturation. Ma et al. revealed that treatment with an anti-CD40 antibody prevented lymphoma development in a cord blood model when mice were infected with the LMP1-deficient EBV strain [11]. LMP1 is an EBV oncogene that mimics a constitutively active CD40 [61,62]. Anti-CD40 antibody treatment also decreases the number of lymphoma cases produced by wild-type EBV [11]. Our findings support the current approach to target the CD40/CD40L pathway in a transplant setting. The CD40 blocking antibody is a potential agent to prevent allograft rejection, which has profoundly affected graft survival in non-human primates [63] and human clinical trials [64]. CD40 blocking antibody could serve a dual purpose by increasing graft survival while preventing EBV-PTLD. Immunosuppressed patients might also benefit from monitoring their serum level of the chemokine CXCL13. CXCL13 is the agonist of Tfh defining marker CXCR5, and its serum levels have been proven to predict B-cell NHL risk in HIV+ individuals [65]. A pediatric case–control study demonstrated that serum CXCL13 level was significantly higher in PTLD patients than in healthy controls [66]. In their research, the elevated CXCL13 serum levels in the pediatric PTLD patients were detected up to two years before the development of the PTLD [66]. Since the survival of mice in the Tfh-depletion group in our study did not reach statistical significance, it stands to reason that other CD4+ Th cells subsets may permit EBV–infected B-cells survival in SCID mice as CD4+ Th cells are more plastic and play dual roles [67].

Treg cells have immunosuppressive properties that prevent the development of autoimmune diseases [68,69]. However, they also hinder antitumor responses and provide an immunosuppressive tumor microenvironment [70]. The high frequency of Treg in peripheral blood is associated with unfavorable outcomes in patients with a broad spectrum of malignancies [71,72]. Furthermore, in vitro depletion of Treg enhances EBV-specific gamma interferon expression by CTLs [73]. Li and others reported that EBV-positive tumor cells recruit Tregs to the tumor site and contribute toward an immune suppressive tumor microenvironment [20]. Treg cells diminish the activity of EBV-specific CTL response [74], and Treg signature cytokines (TGF-beta and IL-10) are associated with the development of EBV-associated diseases [75,76,77]. In our study, the depletion of Treg significantly improved survival (Figure 5c). The GSEA analysis showed that the TGF-beta hallmark gene set was enriched among HI donors (Figure 4b) compared to NI donors. TGF-beta is critical for driving FOXP3 expression, a transcription factor involved in the initiation and maintenance of Treg cells [78]. Indeed, TGFBR2 and FOXP3 transcripts were significantly over-expressed in HI donor’s CD4+ cells (Figure 4c). Others reported that the in vivo neutralization of TGF-beta results in endogenous EBV-specific CD8 activation and expansion and prevents EBV-LPD [76].

Our phenotypic analysis of donor PBMCs revealed a high frequency of MDSCs in PBMCs from HI donors. We hypothesized that MDSCs inhibitory mechanism might effectively block EBV-specific T-cell expansion and play a role in the efficiency of lymphomagenesis in HI donors. Indeed, the in vitro T-cell co-cultures with myeloid cells showed reduced memory CTL proliferation among HI donors. However, when cultured alone, T-cells from NI and HI donors have similar proliferation rates to LCL stimulation (T-cell: MDSC ratio 1:0) (Appendix A). A recent study showed that patients with Chronic Active Epstein–Barr virus (CAEBV) possess significantly higher absolute numbers of circulating MDSCs [79]. Several cytokines play a part in MDSC’s suppressive role in T-cell function and proliferative effector cell activity. This is the topic of ongoing research with a larger cohort of classified donors to help us define the nature of MDSC’s in EBV-LPD.

## 5. Conclusions

Strategies to better predict which individuals are at the highest risk for the development of EBV-LPD represent an unmet need. Our work here provides the rationale for developing prospective clinical trials evaluating EBV-LPD risk profiles for patients undergoing solid organ transplantation and prospective evaluation of their CD4+ Th subset immunophenotyping. Ultimately, results from this work may apply to individuals with primary (genetic) or acquired (HIV) immune deficiency. Future studies on a larger cohort of classified donors (HI and NI donors) are underway to help characterize the role of immune cell subsets in promoting or preventing EBV-LPD.

## Figures and Tables

**Figure 1 cancers-15-03046-f001:**
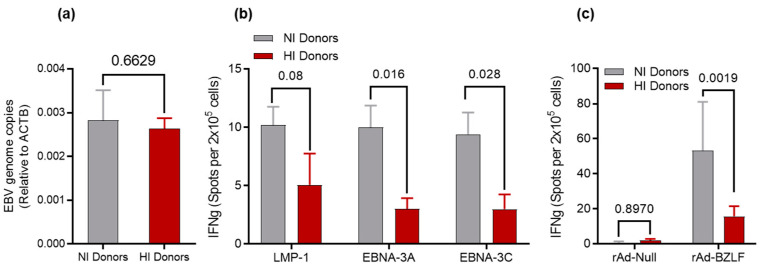
(**a**) Quantitative PCR: DNA was isolated from FACS-sorted memory B-cells to measure EBV-genome copy number. No difference in the EBV copy number between NI and HI donors; IFNγ Elispot: (**b**) PBMCs from NI and HI donors stimulated with pepmix for the EBV latent antigens: EBNA-3A, EBNA-3C, and LMP1; (**c**) PBMCs from NI and HI donors stimulated with adenovirus encoding recombinant BZLF1 (rAd-BZLF1) and the empty adenovirus vector (rAd-null).

**Figure 2 cancers-15-03046-f002:**
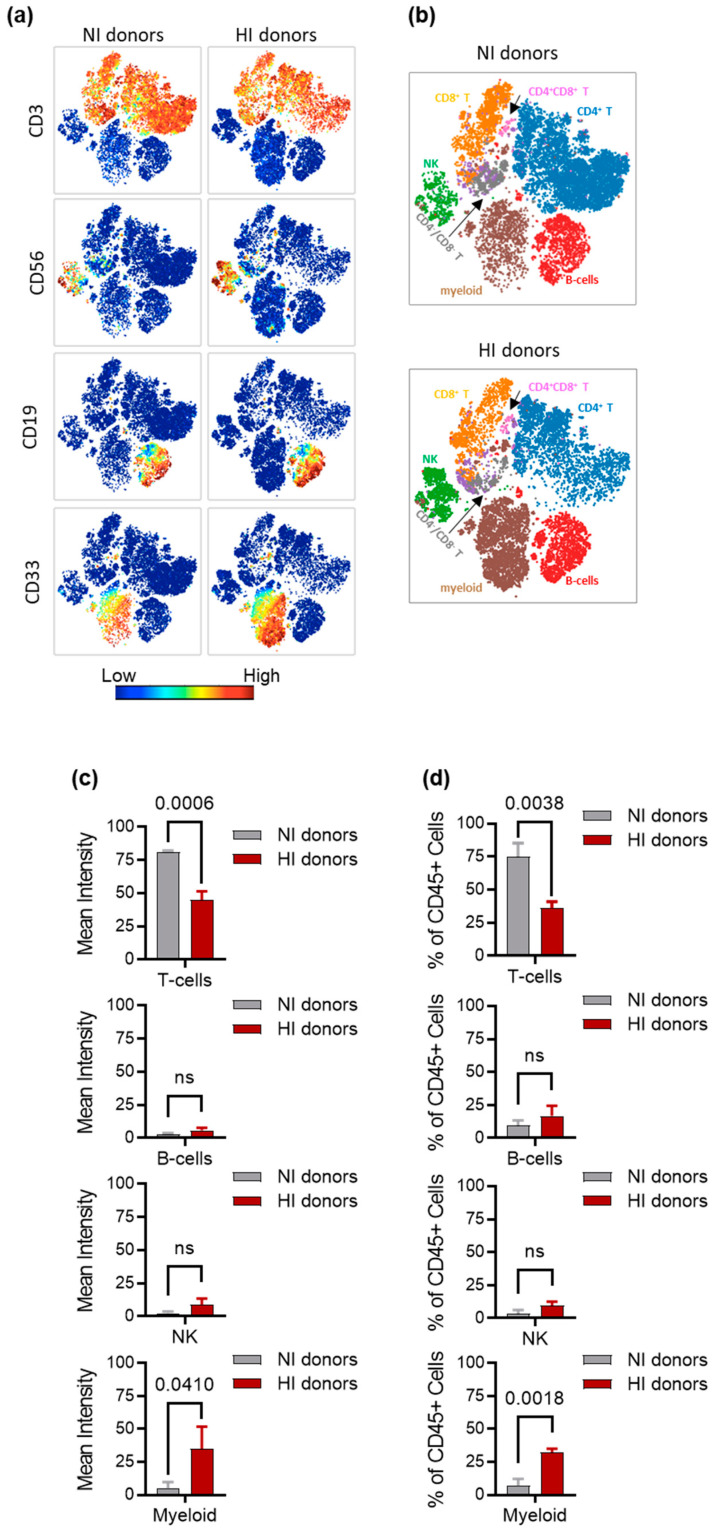
NI and HI donors’ PBMCs phenotype: (**a**) PBMCs from NI donors and HI donors were examined using viSNE clustering. Live (Ir-191-positive) CD45 cells were selected for the clustering, and the viSNE immunome maps were colored by the intensities of individual lineage marker expression (CD3, CD56, CD19, CD33); (**b**) overlaid viSNE maps on gated populations; (**c**) mean intensity levels of markers defined in Figure 2a; (**d**) relative proportions of cell populations in CD45 gate.

**Figure 3 cancers-15-03046-f003:**
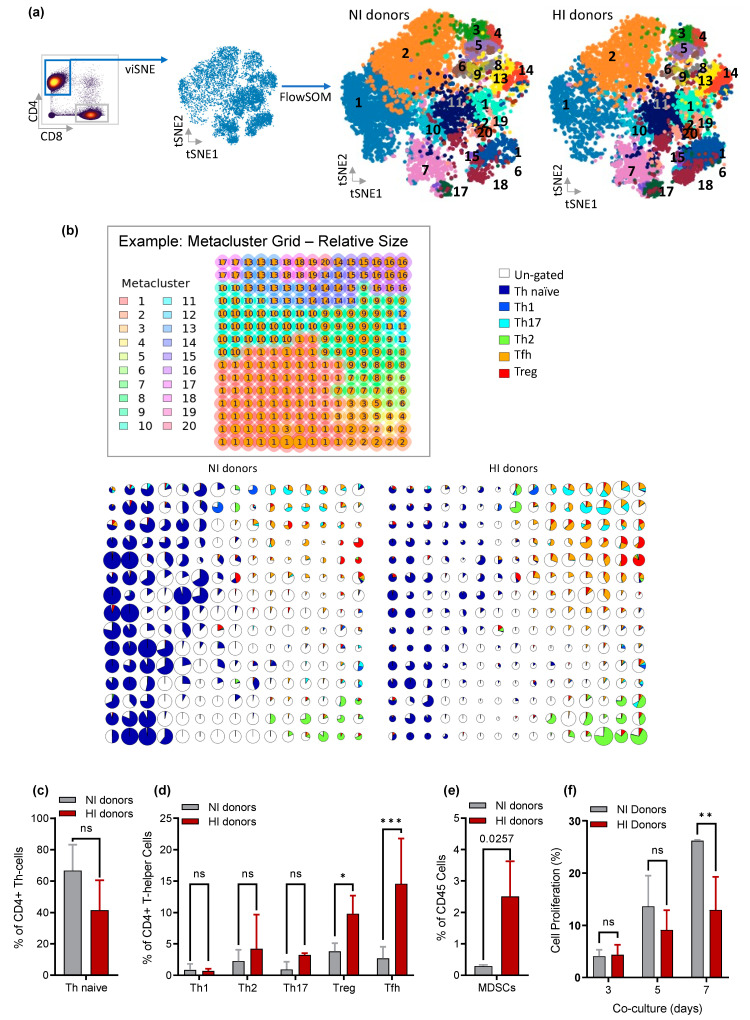
(**a**) Overview of the CD4+ Th cell analysis steps; (**b**) FlowSOM clustering with the percentage of cells falling in the original manual gates; (**c**) the frequency of naïve Th cells: (**d**) the frequency of CD4+ Th subsets: T helper 1 (Th1), T helper 2 (Th2), T helper 17 (Th17), regulatory T cells (Treg) and T follicular helper (Tfh), between NI and HI donors; (**e**) frequency of MDSCs of each group; (**f**) percentage of CD8+ cytotoxic T-cell (CTL) proliferation after activation and culture with autologous myeloid cells. (*) = 0.0307, (**) = 0.0250, and (***) = 0.0002.

**Figure 4 cancers-15-03046-f004:**
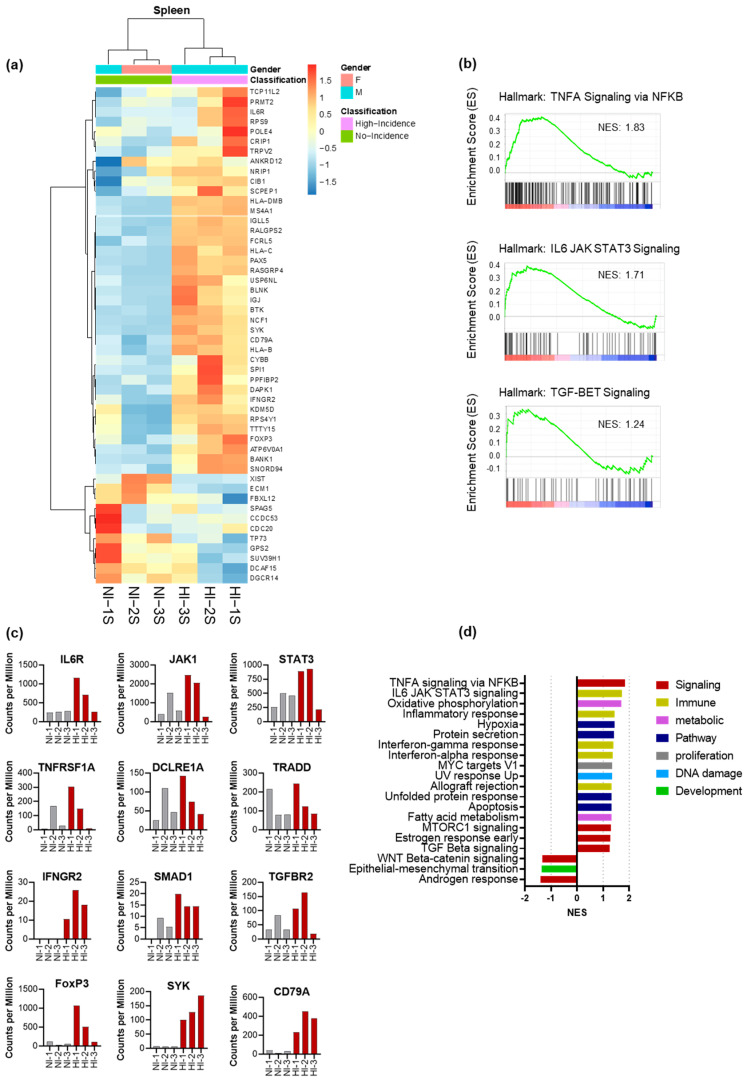
Ex vivo CD4+ Th cell gene signature: (**a**) Heat map showed that the CD4 Transcriptome program is broadly different between NI and HI donors. The top genes are relatively overexpressed, and the bottom genes are relatively under-expressed in HI donor CD4+ Th cells; (**b**) Gene set enrichment analysis (GSEA) identified many crucial pathways for lymphoma growth and survival; (**c**) Representative plots of counts per million in comparison between HI and NI donor’s gene expression; (**d**) Overlapping gene sets between HI and NI donor DEGs and the Molecular Signatures Database (MSigDB) gene sets (Hallmark). The HI donor upregulated genes sets were significant at the FDR < 25%. The downregulated gene set did not reach the FDR < 25% threshold.

**Figure 5 cancers-15-03046-f005:**
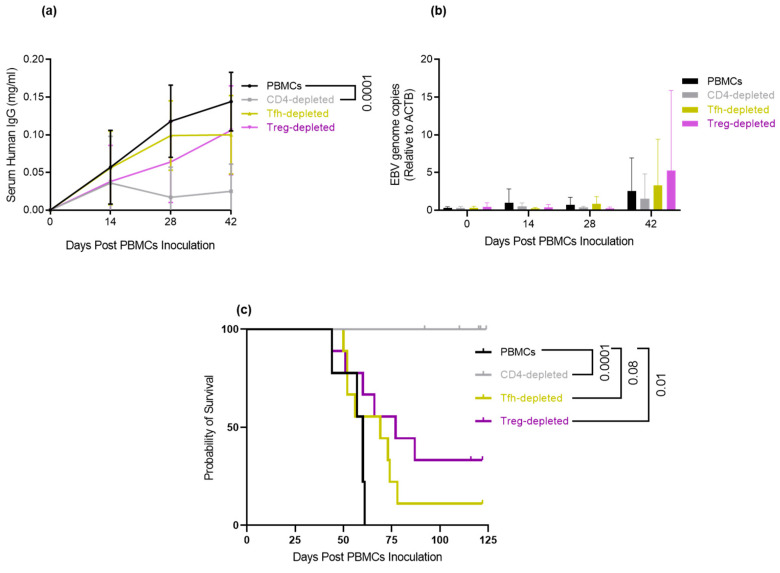
CD4+ T-cell subsets depletion experiment: (**a**) ELISA results of human IgG level in mouse serum collected from mice engrafted with NI or HI donor mock-depleted PBMCs, CD4-depleted PBMCs, Tfh-depleted PBMCs, or Treg-depleted PBMCs. Samples were collected at day 0, 14, 28, and 42 after cell inoculation. (**b**) Quantitative PCR of EBV-genome copy number on mouse serum samples described in Figure 5a. None of the differences were significant; (**c**) Log-rank test for survival.

## Data Availability

All data pertaining to this study are contained within this article.

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
