# Peer review of "Follicular Helper and Regulatory T Cells Drive the Development of Spontaneous Epstein–Barr Virus Lymphoproliferative Disorder"

_cancers, 2023, doi:10.3390/cancers15113046_

Round 1

Reviewer 1 Report

This study describes an original investigation in Immunodeficient mice engrafted with blood cells from EBV+ individuals . A transcriptomic analysis of CD4+ T cells from ex-vivo high incidence donor PBMC revealed amplified cytokine and inflammatory gene signatures. HI vs. NI donors showed a marked reduction in IFN production to EBV latent and lytic antigen stimulation. They observed abundant myeloid-derived suppressor cells in HI donor PBMC that decreased CTL proliferation in co-cultures with autologous EBV+ lymphoblasts.

Their findings identify potential biomarkers that may identify individuals at risk for EBV-LPD and suggest possible strategies for prevention.

The major criticism is the choice of the BZLF1 encoded protein (also called ZEBRA). Why were other lytic cycle antigens not chosen? It is understandable that BZLF1 was chosen by the authors, in view of their previous publications (ie ref 27). The choice of BZLF1 could be better explained in the text, notably by the fact that BZLF1 is a lytic cycle protein involved in tumour progression (Germini D, Sall FB, Shmakova A, Wiels J, Dokudovskaya S, Drouet E, Vassetzky Y. Oncogenic Properties of the EBV ZEBRA Protein. Cancers (Basel). 2020 Jun 5;12(6):1479. doi: 10.3390/cancers12061479. PMID: 32517128; PMCID: PMC7352903.)

Certainly the authors should better validate the choice of this relevant antigen.

The major critic of this article is the choice of BZLF1 encoded protein (ZEBRA transactivator of EBV) as antigen

Reviewer 2 Report

Manuscript Nr.: cancers-2358730

Ahmed et al., “Follicular Helper and Regulatory T cells drive the development of spontaneous Epstein-Barr virus lymphoproliferative disorder

The authors demonstrate that 20% of healthy PBMC donors allow for the development of Epstein Barr virus (EBV) associated lymphoproliferative disease (LPD) after their PBMC transfer into SCID mice. This seems to mimic the frequency of post-transplant lymphoproliferative disease (PTLD) development in many clinical centers, even so it seems difficult to compare the immunosuppressed immune compartments of transplant patients with PBMCs of healthy EBV carriers. Nevertheless, the authors reveal differences between the PBMC donors that reliably cause LPDs after their PBMC transfer (HI) compared to those that do not elicit LPDs (NI). NI have higher levels of EBV specific T cells and T cells overall, while HI have higher frequencies of myeloid cells, myeloid-derived suppressor cells (MDSCs), follicular helper T cells (Tfh) and regulatory T (Treg) cells. The authors argue that MDSCs, Tfh and Treg attenuate EBV specific T cell responses that prevent LPDs in NI. Depletion of Tregs in HI PBMCs prior to transfer decreases LPD development. The authors suggest that these leucocyte populations could be used to distinguish transplant patients with or without risk of PTLD development.

This is an interesting proposition. However, the functional studies on MDSCs, Tfh and Treg are rather unclear and should be improved.

Major comments:

1.   The study is based on the different outcomes of PBMC transfer into SCID mice. Therefore, the classification into HI and NI donors and for classifying a mouse as LPD carrier should be better described. What are the criteria for LPD development after PBMC transfer?

2.   Which of the implicated leucocyte populations survive after PBMC transfer and could continue to shape LPD development? Do MDSCs survive after PBMC transfer until the sacrifice time points of 4 weeks after injection? Do Tfh and Treg expand in number and frequency comparing the transferred PBMC preparation and the splenocytes after four weeks of PBMC transfer into SCID mice?

3.   The in vitro studies on the suppressive function of MDSCs are also unclear because the difference in T cell proliferation could originate from a higher frequency of EBV specific T cells in NI donors or a stronger immune suppressive MDSC population in HI donors. Do T cells from HI donors proliferate in response to LCL stimulation similarly to NI donors if myeloid cells are not added? Do MDSCs from HI donors produce soluble factors that would also suppress T cell proliferation of NI donor PBMCs?

4.   No data is provided on the efficacy of the T cell subset depletions. Does CD25 depletion primarily target FoxP3 positive CD4+ T cells in the authors’ PBMC preparations and how many CXCR5 positive CD4+ T cells are in HI donors’ PBMCs?

5.   The H&E staining in figure 5C does not seem to be easily interpretable and no quantification is provided. Staining for an EBV gene produce, such as EBNA2, and quantification across several white pulp regions would need to be done to argue for differences.

Minor comments:

1.   While the methods correctly state that the follicular helper T cells were depleted with a CXCR5 targeting antibody, the results suggest depletion with a CXCR3 specific antibody. This should be clarified. Most likely this is a typo and CXCR5 should be stated in both methods and results.

2.   Figure legend 5A should probably state CD4-depleted and not -deplated.

Round 2

Reviewer 2 Report

The authors demonstrate that 20% of healthy PBMC donors allow for the development of Epstein Barr virus (EBV) associated lymphoproliferative disease (LPD) after their PBMC transfer into SCID mice. This seems to mimic the frequency of post-transplant lymphoproliferative disease (PTLD) development in many clinical centers, even so it seems difficult to compare the immunosuppressed immune compartments of transplant patients with PBMCs of healthy EBV carriers. Nevertheless, the authors reveal differences between the PBMC donors that reliably cause LPDs after their PBMC transfer (HI) compared to those that do not elicit LPDs (NI). NI have higher levels of EBV specific T cells and T cells overall, while HI have higher frequencies of myeloid cells, myeloid-derived suppressor cells (MDSCs), follicular helper T cells (Tfh) and regulatory T (Treg) cells. The authors argue that MDSCs, Tfh and Treg attenuate EBV specific T cell responses that prevent LPDs in NI. Depletion of Tregs in HI PBMCs prior to transfer decreases LPD development. The authors suggest that these leucocyte populations could be used to distinguish transplant patients with or without risk of PTLD development.

The revision addressed most of my comments. Namely, characteristics of the authors LPD classification were now more clearly described. Moreover, it was clarified that mostly lymphocytes survive the four weeks after PBMC transfer, suggesting limited influence of MDSCs in vivo. Supplemental data were provided that NI and HI donor PBMCs proliferate similarly to LCL stimulation in the absence of MDSCs. A statement was also included that CD25 depletion mainly affected FoxP3 positive cells, even so no quantitative assessments were provided. Finally, no additional histochemistry data could be provided due to lack of tissue from the described experiments. Therefore, I would move figure 5C to the supplement.   
